

# Small tropical islands with dense human population: differences in water quality of near-shore waters are associated with distinct bacterial communities

Hauke F. Kegler[1,2], Christiane Hassenrück[1], Pia Kegler[3], Tim C. Jennerjahn[1], Muhammad Lukman[4], Jamaluddin Jompa[4] and Astrid Gärdes[1]

[1] Department of Biogeochemistry and Geology, Leibniz-Centre for Tropical Marine Research, Bremen, Germany
[2] Faculty of Biology and Chemistry (FB2), University of Bremen, Bremen, Germany
[3] Department of Ecology, Leibniz-Centre for Tropical Marine Research, Bremen, Germany
[4] Department of Marine Science, Universitas Hasanuddin, Makassar, Indonesia

Corresponding author
Hauke F. Kegler,
hauke.kegler@gmail.com

## ABSTRACT

Water quality deterioration caused by an enrichment in inorganic and organic matter due to anthropogenic inputs is one of the major local threats to coral reefs in Indonesia. However, even though bacteria are important mediators in coral reef ecosystems, little is known about the response of individual taxa and whole bacterial communities to these anthropogenic inputs. The present study is the first to investigate how bacterial community composition responds to small-scale changes in water quality in several coral reef habitats of the Spermonde Archipelago including the water column, particles, and back-reef sediments, on a densely populated and an uninhabited island. The main aims were to elucidate if (a) water quality indicators and organic matter concentrations differ between the uninhabited and the densely populated island of the archipelago, and (b) if there are differences in bacterial community composition in back-reef sediments and in the water column, which are associated with differences in water quality. Several key water quality parameters, such as inorganic nitrate and phosphate, chlorophyll $a$, and transparent exopolymer particles (TEP) were significantly higher at the inhabited than at the uninhabited island. Bacterial communities in sediments and particle-attached communities were significantly different between the two islands with bacterial taxa commonly associated with nutrient and organic matter-rich conditions occurring in higher proportions at the inhabited island. Within the individual reef habitats, variations in bacterial community composition between the islands were associated with differences in water quality. We also observed that copiotrophic, opportunistic bacterial taxa were enriched at the inhabited island with its higher chlorophyll $a$, dissolved organic carbon and TEP concentrations. Given the increasing strain on tropical coastal ecosystems, this study suggests that effluents from densely populated islands lacking sewage treatment can alter bacterial communities that may be important for coral reef ecosystem function.

# INTRODUCTION

Coral reef systems in close vicinity to highly populated areas are often affected by land-based activities, e.g., intensive farming and coastal development, which can lead to additional nutrient release to near-shore waters (*Bell, 1992*; *Gardner et al., 2003*; *Lapointe, Barile & Matzie, 2004*; *Wolanski, Martinez & Richmond, 2009*; *Reopanichkul et al., 2010*). This increase of inorganic and organic nutrients can lead to the proliferation of macroalgae and stimulation of phytoplankton growth, with a subsequent increased light attenuation and sedimentation, effectively reducing gross photosynthesis, smothering the corals and preventing recruitment (*Fabricius et al., 2003*; *Fabricius, 2005*). Additional organic matter derived from phytoplankton production can also result in community shifts and an increased activity of the bacterial community with potentially harmful effects for corals, such as an increased prevalence of pathogens or lowered oxygen concentrations (*Kuntz et al., 2005*; *Garren, Smriga & Azam, 2008*).

In organic matter-rich conditions, bacterial community composition can shift from autotrophic to heterotrophic in reef waters (*Meyer-Reil & Köster, 2000*; *Weinbauer et al., 2010*; *De Voogd et al., 2015*), microbial biofilms (*Sawall, Richter & Ramette, 2012*; *Witt, Wild & Uthicke, 2012*), and sediments (*Uthicke & McGuire, 2007*). This shift often occurs alongside an increase in total bacterial cell counts (*Zhang et al., 2007*, *2009*; *Dinsdale et al., 2008*). High bacterial abundance and remineralization increase the oxygen consumption and lead to hypoxia, with potentially fatal consequences for benthic organism (*Kline et al., 2006*). Bacteria also play an important role in biogeochemical cycling and coral reef health, including nutrient cycling and antimicrobial activities (*Azam et al., 1993*; *Azam, 1998*; *Rosenberg et al., 2007*; *Azam & Malfatti, 2007*; *Stocker, 2012*; *Rädecker et al., 2015*). Therefore, even small shifts due to anthropogenically induced eutrophication can further alter nutrient cycling activities, sedimentation and organic matter export, as well as promote coral pathogens (*Bruno et al., 2003*; *Fabricius, 2005*; *Garren, Smriga & Azam, 2008*; *Lyons et al., 2010*).

Water quality in Indonesia has been declining in the past decade, with domestic wastewater, industry, agriculture, and fish farming being the main sources of pollution (*Asian Development Bank, 2016*). The Spermonde Archipelago in southern Sulawesi, Indonesia, including its approximately 150 small islands, is an excellent example for such anthropogenic impacts, as several previous studies revealed (*Renema & Troelstra, 2001*; *Sawall, Richter & Ramette, 2012*; *Plass-Johnson et al., 2015*). Untreated sewage and pollutants from metropolitan Makassar (population: approx. 1.4 million) enter the system directly or via the river Jene Berang, which additionally discharges sediments, suspended particulate matter, and inorganic nutrients from the hinterland (*Renema & Troelstra, 2001*; *Nasir et al., 2016*). The Archipelago has therefore been characterized by a cross-shelf gradient from chlorophyll *a*-rich inshore to oligotrophic offshore waters

(*Renema & Troelstra, 2001*; *Becking et al., 2006*; *Kegler et al., 2017*). Subsequently, deteriorating coral reefs of the fringing islands in the Spermonde Archipelago could put the livelihoods of thousands of fishermen at risk (*Pet-Soede et al., 2001*).

In addition to anthropogenic disturbances on the larger-scale, driven by sewage and riverine input from the mainland and Makassar, many of the small islands are also densely populated and lack proper sewage and waste water treatment facilities. There is some indication that dense human populations inhabiting small islands, such as Spermonde Archipelago, can have profound effects on their physicochemical environment (*White & Falkland, 2010*). For example, on some islands there is now an imbalance between the availability and the demand of freshwater with increasing human populations (*Schwerdtner Máñez et al., 2012*), or a pollution of the freshwater lens (*White & Falkland, 2010*). Therefore, groundwater seepage (*Laws, Brown & Peace, 2004*; *Paytan et al., 2006*) and freshwater runoff can be sources of inorganic and organic nutrients to the fringing reefs. Those nutrients are quickly assimilated and converted into phytoplankton biomass (*Koop et al., 2001*). Consequences are an increased release of dissolved organic carbon (DOC) and transparent exopolymer particles (TEP; *Karl et al., 1998*; *Passow, 2002*) and subsequent aggregation and sedimentation of large particles rich in organic matter (*Kiørboe & Hansen, 1993*; *Cárdenas et al., 2015*).

There is only little information available on bacterial community composition in the Spermonde Archipelago. The studies conducted so far have focused on the diversity of settlement tile biofilms (*Sawall, Richter & Ramette, 2012*) and bacterial communities from different reef habitats, specifically within sponges, as well as on the functional role of the associated bacteria (*Cleary et al., 2015*). A more recent study investigated the abundance of potential pathogens at sites with and without seagrass meadows (*Lamb et al., 2017*). So far, most previous studies, except by *Lamb et al. (2017)*, investigated changes in bacterial community composition along much larger gradients (*Kegler et al., 2017*). This will be the first study in the Archipelago to simultaneously investigate small-scale spatial differences in water quality and bacterial community composition while comparing two islands of different inhabitation status. It aims to elucidate (i) how the inhabitation status of two (one uninhabited and one densely populated) islands affects water quality, including inorganic nutrients, chlorophyll *a*, DOC, and TEP at increasing distance from the islands, and (ii) how those possible differences in water quality may affect bacterial community composition within the reef sediments as well as in the free-living and particle-attached fractions of the water column.

## MATERIAL AND METHODS

### Location and characteristics of the sampling sites

Two transects, each at two islands of similar distance from Makassar, (South Sulawesi, Indonesia), were sampled during the dry south–east monsoon in May 2014 (Fig. 1). One island, Pulau Barrang Lompo, sampled on May 19, is densely populated, with more than 4,000 people inhabiting the 0.2 km$^2$ island, while the uninhabited island Pulau Kodinggareng Keke, sampled on May 29, was chosen as a reference. The two replicate transects were laid out from 25 to 300 m, at depths ranging from app. 50 cm to a maximum of 3 m, across the northern and southern parts of the back-reef area at each

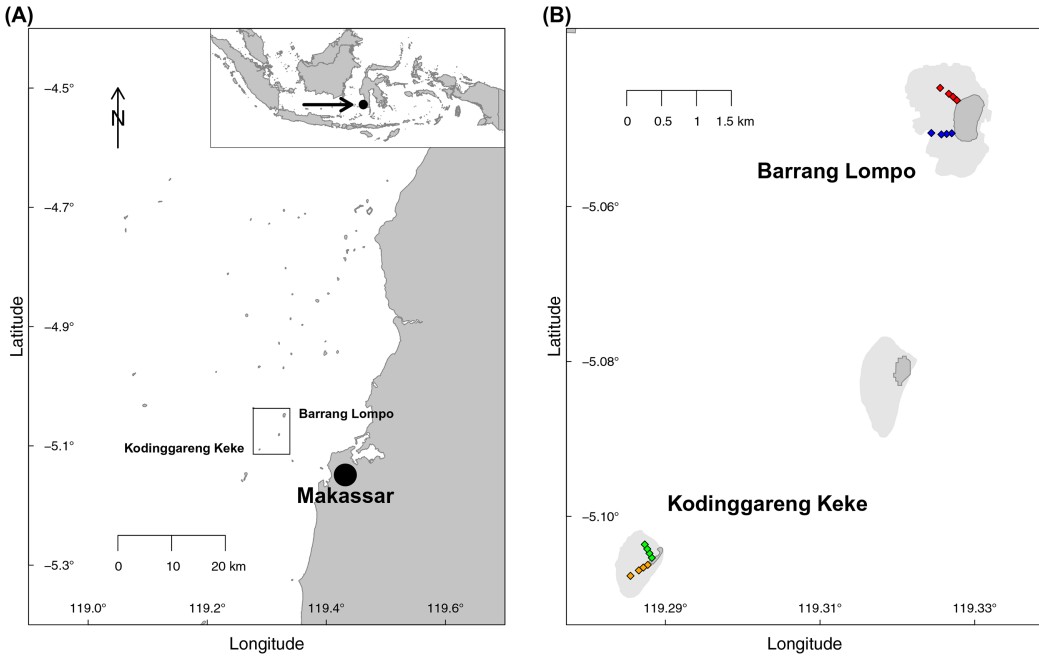

**Figure 1 Map of the sampling area.** (A) Spermonde Archipelago, South Sulawesi, Indonesia. (B) Close-up of the two sampled islands Pulau Barrang Lompo (inhabited) and Pulau Kodinggareng Keke (uninhabited) including the position of the sampling sites along the south and north transects across the back-reef area.

island (Fig. 1). Samples were collected at four transect points (sites) at 25, 75, 150, and 300 m distance from the islands. All sampling was conducted while snorkeling during high tide in the morning hours and completed in one day for each island. Coral reefs and seagrass meadows were present in the vicinity, but care was taken not to take sediment samples directly within those habitats. Temperature (average 31 °C), pH (average 8.08), and salinity (average 32.8) were all within a very narrow range of less than 1% variability across both islands. The required research permits, permit no. 5599/FRP/SM/XI/2013 (Dr. Astrid Gaerdes), 5552/FRP/SM/XII/2013 (Dr. Hauke Kegler), and 176/SIP/FRP/SM/V/2013 (Dr. Pia Kegler) to conduct fieldwork in Indonesia were provided by the Indonesian Ministry of Research, Technology, and Higher Education (RISTEKDIKTI).

## Water quality parameters

Chlorophyll *a* was measured in situ, at depths ranging from app. 50 cm to a maximum of 3 m, with a Eureka Manta 2 multiprobe (Eureka Water Probes, Austin, TX, USA). Additionally, surface water was collected at all sampling points in triplicates, using 5 L high-density polyethylene canisters, washed with 1M HCl and rinsed with distilled water, for determination of inorganic nutrients, DOC, and TEP. From each replicate canister subsamples of 30 mL (DOC) and 50 mL (inorganic nutrients) each were taken. Within 3 h of sampling, all samples were transported to the field station on Pulau Barrang Lompo and kept in the dark until filtration. After filtration through a 0.7 μm pore size GF/F filter (GE Healthcare Bio-Sciences, Pittsburgh, PA, USA), samples for inorganic nutrients (nitrite, nitrate, phosphate, and silicate) were immediately fixed with mercury chloride

and frozen at −20 °C until spectro-photometrical analysis with a Flowsys continuous flow analyzer (Systea, Anagni, Italy). To determine DOC, subsamples were filtered through a 0.45 µm pore size cellulose acetate syringe filter (Sartorius, Goettingen, Germany), acidified with HCl (pH < 2) and frozen at −20 °C until further analysis. DOC concentrations were measured by high-temperature oxic combustion using a TOC-VCPH analyzer (Shimadzu, Mandel, Ontario, Canada). Hansell artificial seawater standards (Hansell Laboratory RSMAS, University of Miami, Miami, FL, USA) served as a quality control, while ultrapure water was used for the blanks. For TEP analysis the spectrophotometric method first introduced by *Passow & Alldredge (1995)* was used, with an updated protocol by *Engel (2009)*. Briefly, this method relates the adsorption of a dye to the weight of polysaccharides filtered on 0.4 µm pore size polycarbonate filters (GE Healthcare Bio-Sciences, Pittsburgh, PA, USA). A calibration curve was prepared using the reference polysaccharide Gum Xanthan from *Xanthomonas campestris* cultures.

## DNA extraction and Illumina sequencing

To separate bacterioplankton into two size fractions, representing "particle-attached" and "free-living" bacteria following the classification of *Crump, Armbrust & Baross (1999)*, a 1 L subsample from each transect point was filtered sequentially using 3 and 0.2 µm Whatman Nuclepore polycarbonate filters (GE Healthcare Bio-Sciences, Pittsburgh, PA, USA). We applied the protocol established by *Boström et al. (2004)* for DNA extraction from water column samples. Surface sediment was collected from the uppermost 1 cm of sediment at each transect point. Sediment samples were collected in 2.0 mL tubes in situ (Eppendorf, Hamburg, Germany), stored at −20 °C until extraction and shipped frozen on ice. Extraction was carried out at facilities of the Leibniz Centre for Tropical Marine Research in Bremen, Germany, using the PowerSoil™ DNA Isolation Kit (MO BIO Laboratories, Carlsbad, CA, USA) with modification of the following two steps of the protocol: (1) we did not incubate for 5 min at 4 °C, but centrifuged the sample directly after the addition of S2, and (2) we used 50 µL of elution buffer instead of 100 µL. All extracted DNA samples were sequenced by LGC Genomics (Berlin, Germany) using an Illumina MiSeq V3 reagent kit (Illumina Inc., San Diego, CA, USA). The 16S rRNA primers 341F (5′-CCTACGGGNGGCWGCAG-3′) and 785R (5′-CTACCAGGGTATCTAATCC-3′) were used, targeting the V3–V4 hypervariable region (*Klindworth et al., 2013*). For the sequencing library preparation, PCR reactions were run with approximately 5 ng of DNA template, 15 pmol of each forward and reverse primer, and 1× MyTaq buffer, consisting of 1.5 units MyTaq DNA polymerase (Bioline Reagents Ltd., London, UK) and 2 µL of BioStabII PCR Enhancer (Sigma-Aldrich Chemie GmbH, Munich, Germany), in a total reaction volume of 20 µL. After an initial denaturation step at 96 °C for 2 min, PCRs were run with 30 cycles of 15 s at 96 °C, 30 s at 50 °C, and 90 s at 70 °C.

Demultiplexed and primer-clipped sequences provided by LGC Genomics (available at the European Nucleotide Archive, accession number PRJEB18570) were used for the analysis with DADA2 version 1.4 (*Callahan et al., 2016*) to generate operational taxonomic units (OTU) based on unique sequence variants. Briefly, sequences were

quality filtered at a maximum expected error rate of three and a minimum length of 245 and 205 bp for forward and reverse reads, respectively. Error learning and denoising were run with default settings pooling all sequences across samples. After denoising, forward and reverse reads were merged with a minimum overlap of 10 bp and without allowing mismatches in the overlap region. Only merged sequences with a length between 380 and 430 bp were considered for further analysis. Chimera removal was conducted using default settings. Then, sequences were submitted to SILVAngs (https://www.arb-silva.de/ngs/; date accessed 02.08.2017) for taxonomic classification using the SILVA ribosomal RNA gene database version 128 as reference (*Quast et al., 2013*). OTUs with a sequence similarity of less than 93% to the reference database, OTUs unclassified on phylum level, and OTUs affiliated with eukaryotic, archaeal, mitochondrial, or chloroplast sequences were removed from the data set. R and bash scripts for sequence processing are available as Supplementary Material. DADA2 was run in R version 3.4.1 (*R Core Team, 2017*).

## Statistical analysis

A principle component analysis (PCA) was conducted to visualize changes in water quality parameters among samples.

Differences in individual water quality parameters between islands and with increasing distance from the island were assessed using general linear mixed models (GLMM) with transect as random factor (*Kuznetsova, Brockhoff & Christensen, 2016*). Prior to analysis, values were log-transformed to meet the assumption of normality.

Alpha diversity of the bacterial communities was assessed based on the Inverse Simpson Index (*Hill, 1973*) after randomly rarefying the data set repeatedly to the minimum library size (964 sequences). Differences in alpha diversity between islands and correlations with water quality parameters were assessed using nonparametric Wilcoxon tests and Spearman correlations, respectively.

Nonmetric multidimensional scaling (NMDS), based on Bray–Curtis dissimilarities of relative sequence abundances, was conducted to visualize patterns in the bacterial community composition for each habitat. Correlations of patterns in community composition and water quality parameters were mapped onto the NMDS plot using the R function *envfit* of the vegan package (*Oksanen et al., 2016*). Analysis of similarity (ANOSIM) was used to assess the separation of bacterial communities between different habitats, and between the two islands within the same habitat based on Bray–Curtis dissimilarity coefficients (*Clarke, 1993*).

Redundancy analyses (RDA; *Legendre & Legendre, 1998*) were run to evaluate the ability of the inhabitation status of the islands (inhabited, uninhabited) to explain the variation in bacterial community composition. Prior to RDA, OTUs that only occurred once at either of the islands were removed from the data set. Furthermore, sequence counts were centered log ratio (clr)-transformed using the R function *aldex.clr* of the ALDEx2 package (*Fernandes et al., 2014*) using median values of 128 Monte-Carlo instances. To compare the explanatory power of inhabitation status and water quality parameters, additional RDA models were constructed with water quality parameters as predictors. Separate models were run with either inhabitation status or water quality parameters as predictors.

Forward model selection was used after checking for variance inflation to determine the water quality parameters to be included in the RDA models. In cases where more than one water quality parameter was selected, pure effects were also tested accounting for the variation explained by the other factors in the model.

Variance inflation factors (VIFs) of the individual water quality parameters were used to check for collinearity among predictors. None of the parameters in any of the RDA models displayed VIFs larger than 10. The water quality parameters best suited for the model were further determined using forward model selection based on the minimum Akaike information criterion. The adjusted $R^2$ is provided as goodness-fit-statistic.

Random forest analysis (*Liaw & Wiener, 2002*) was conducted to identify the OTUs that are most important for differentiating bacterial communities from the inhabited and the uninhabited island. As importance criterion mean decrease in accuracy of the random forest models was selected. For each habitat, 10,001 random trees were generated. Model significance and accuracy were assessed using permutation tests and leave-one-out cross-validation, respectively, following the tutorial provided on https://github.com/LangilleLab/microbiome_helper/wiki/Random-Forest-Tutorial (date accessed 09.08.2017).

For the statistical analysis we used the following R version and software packages: R software, version 3.3.2 (*R Core Team, 2017*) and R-Studio, version 1.0.153 (*RStudio Team, 2016*), "vegan" (*Oksanen et al., 2016*) for PCA and RDA, "lmerTest" (*Kuznetsova, Brockhoff & Christensen, 2016*) for the GLMM, "mada" (*Doebler & Holling, 2015*) for the calculation of confidence intervals of Spearman correlation coefficients, "iNEXT" (*Hsieh, Ma & Chao, 2016*) for the rarefaction analysis as well as "caret" (*Kuhn, 2017*), "rfUtilities" (*Jeffrey & Murphy, 2017*), and "randomForest" (*Liaw & Wiener, 2002*) for the random forest analysis. Additional packages used within R were "ALDEx2" (*Fernandes et al., 2014*), "ape" (*Paradis, Claude & Strimmer, 2004*), "car" (*Fox, Friendly & Weisberg, 2013*), "gplots" (*Warnes et al., 2016*), "permute" (*Simpson, 2016*), "plyr" (*Wickham, 2011*), and "reshape" (*Wickham, 2007*). Throughout the manuscript concentrations of water quality parameters are given as arithmetic mean ± standard deviation. The R scripts for the statistical data analysis as well as figure generation are available as Supplementary Material.

## RESULTS

### Water quality parameters

Based on GLMMs, all measured water quality parameters, except silicate, were significantly higher at the inhabited than at the uninhabited island. Additionally, we observed a decrease from the sampling point closest to the island to the reef crest that was also statistically significant for all parameters, except silicate and DOC (Fig. 2; Table 1; Table S1). However, in case of DOC as well as inorganic phosphate and chlorophyll *a*, we further detected a significant interaction between island and distance from island, indicating that the trend with increasing distance was different between the two islands (Table 1). The highest $NO_x^-$ (Fig. 2A) and inorganic phosphate (Fig. 2B) concentrations (0.59 ± 0.01 and 0.18 ± 0.01 μmol L$^{-1}$, respectively) were measured on the southern transect of the inhabited island, compared to highest concentrations of 0.16 ± 0.01 and

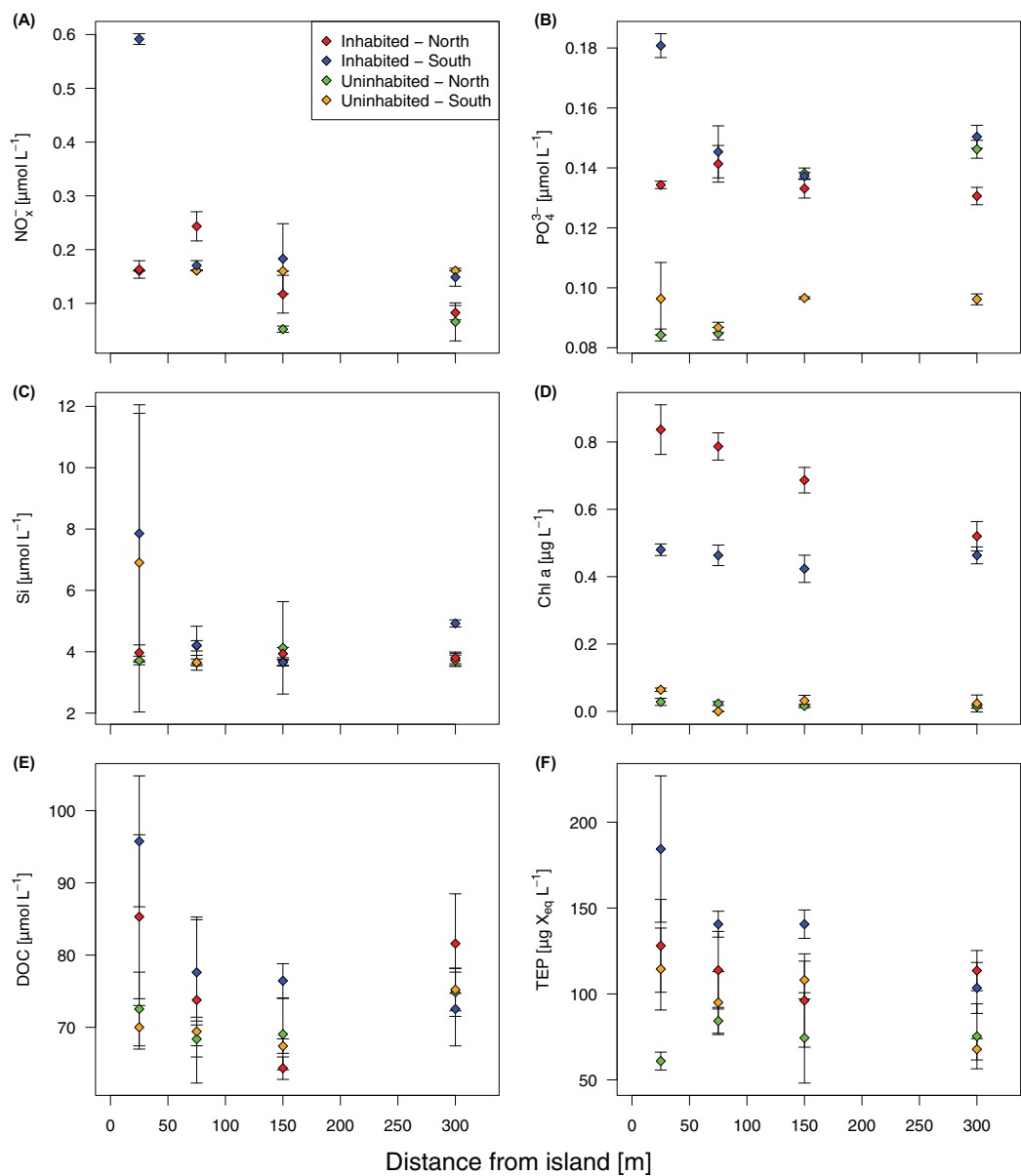

**Figure 2 Water quality parameters at the inhabited and uninhabited island.** (A) Nitrite/nitrate ($NO_x^-$). (B) Phosphate ($PO_4^{3-}$). (C) Silicate (Si). (D) Chlorophyll $a$ (Chl $a$). (E) Dissolved organic carbon (DOC). (F) Transparent exopolymer particles (TEP). Error bars depict standard deviation.

$0.14 \pm 0.01$ $\mu$mol L$^{-1}$, respectively, at the uninhabited island. At both transects, north and south, chlorophyll $a$ concentrations were higher at the inhabited than at the uninhabited island (Fig. 2D). Concentrations of the two different classes of organic matter, DOC (Fig. 2E) and TEP (Fig. 2F), were also higher at the inhabited island. The lowest DOC concentrations ($67.40 \pm 1.00$ $\mu$mol L$^{-1}$) were measured on the southern transect of the uninhabited island, while the highest concentration ($95.76 \pm 9.06$ $\mu$mol L$^{-1}$) was measured at the southern transect of the inhabited island. TEP concentrations at the inhabited island were approximately three times higher ($184.36 \pm 42.59$ compared to

**Table 1 Differences in water quality between islands and with increasing distance from the island, as well as the interaction of these two factors, assess by GLMMs.**

| Parameter[a] | Island | | Distance | | Interaction | |
|---|---|---|---|---|---|---|
| | F | p-Value | F | p-Value | F | p-Value |
| $NO_x^-$ | 10.178 | **0.003** | 28.985 | **<0.001** | 2.099 | 0.155 |
| $PO_4^{3-}$ | 66.904 | **<0.001** | 4.859 | **0.033** | 14.649 | **<0.001** |
| Si | 0.659 | 0.421 | 2.410 | 0.128 | 0.001 | 0.977 |
| Chl $a$ | 322.943 | **<0.001** | 7.753 | **0.008** | 4.133 | **0.048** |
| DOC | 14.239 | **<0.001** | 0.412 | 0.525 | 5.272 | **0.026** |
| TEP | 19.621 | **<0.001** | 8.406 | **0.006** | 0.421 | 0.520 |

Notes:

p-Values defined as significant at a threshold of 0.05 are highlighted in bold.

[a] $NO_x^-$, nitrite/nitrate; $PO_4^{3-}$, phosphate; Si, silicate; Chl $a$, chlorophyll $a$; DOC, dissolved organic carbon; TEP, transparent exopolymer particles.

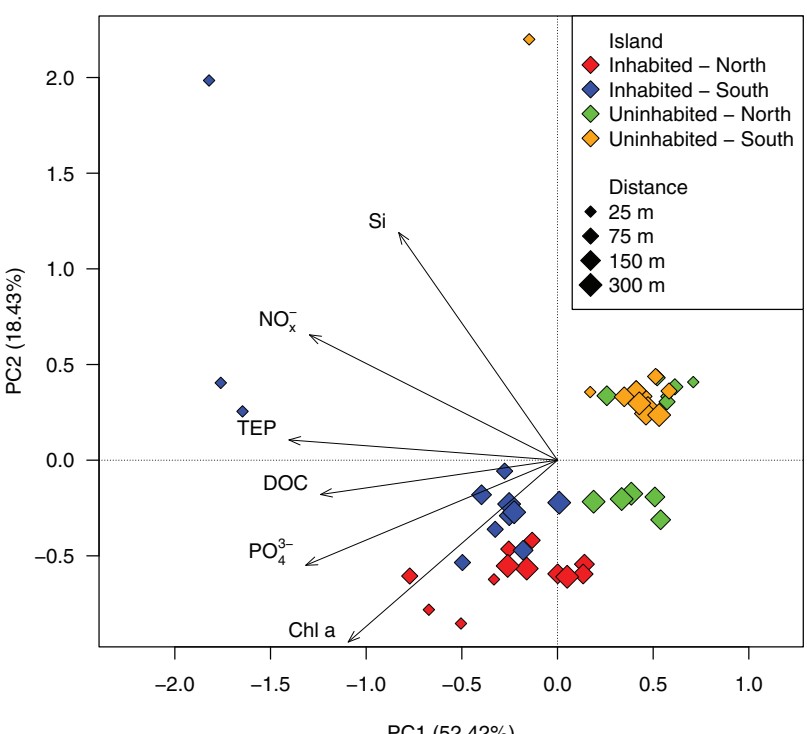

**Figure 3 PCA ordination of the measured water quality parameters at the inhabited and uninhabited island.** PC1 and PC2 explained 52.42% and 18.43% of the total variation, respectively. $NO_x^-$, nitrite/nitrate; $PO_4^{3-}$, phosphate; Si, silicate; Chl $a$, chlorophyll $a$; DOC, dissolved organic carbon; TEP, transparent exopolymer particles.

$67.78 \pm 6.59$ µg Xeq × L$^{-1}$ at the uninhabited island). The first two principal components of the PCA (Fig. 3; Table S2) accounted for 70.8% of the variation in water quality parameters among islands. The sampling sites from the inhabited island separated from those of the uninhabited one by the first principal component. Those differences along the first principal component were mainly driven by TEP, $PO_4^{3-}$, $NO_x^-$, and DOC (Table S2). Differences along the second principal component were driven by chlorophyll $a$ and

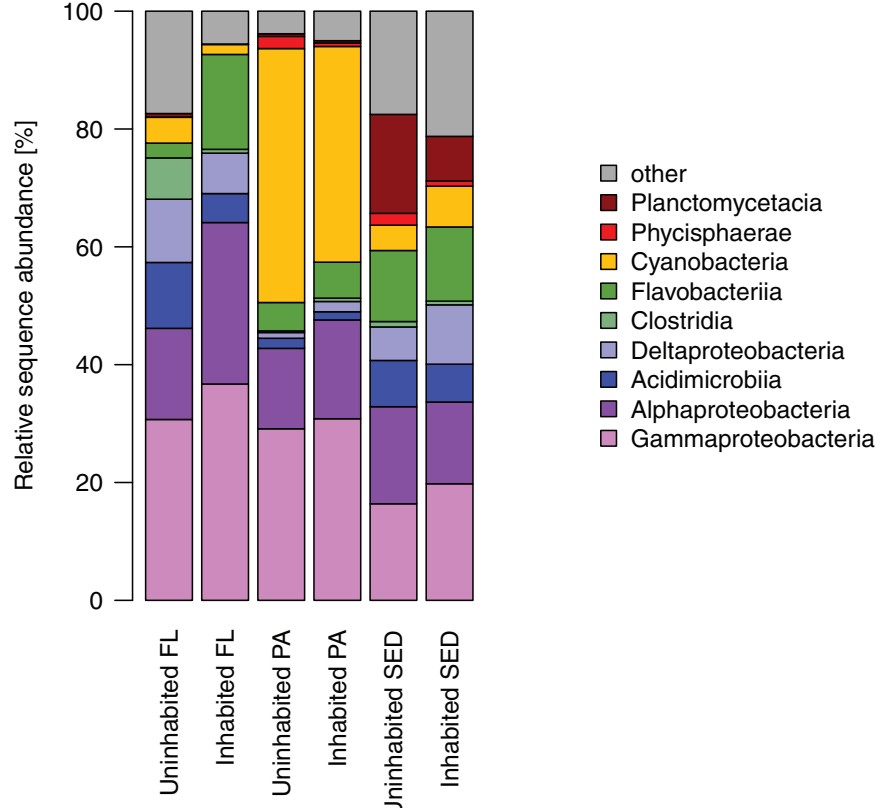

**Figure 4 Relative sequence abundance of bacterial classes in each sampled habitat at the inhabited island and uninhabited island.** FL, free-living bacterial communities of the water column (>0.2 μm); PA, particle-attached bacterial communities of the water column (>3 μm); SED, bacterial communities in reef sediments.

silicate, separating the northern transect of the inhabited from the southern transect of the uninhabited island.

## Bacterial diversity and community composition

In total, we obtained 488,370 quality-checked sequences with an average 11,099 sequences per sample (ranging from 964 to 49,329), corresponding to 3,834 denoised unique sequence variants (OTUs). After rarefaction, the number of OTUs per sample ranged from 125 to 600. Highest average OTU counts (532) were obtained from the sediments, compared to 219 and 180 from the particle-attached and free-living fraction of the water column, respectively. Average inverse Simpson indices were also highest in samples obtained from the sediment, followed by the free-living and particle-attached fraction of the water column (Figs. S1 and S2; Table S3).

   Most of the sequences obtained from the sediment bacterial communities were affiliated with *Gamma-* and *Alphaproteobacteria*, as well as *Flavobacteriia*, *Planctomycetacia*, and *Deltaproteobacteria* (Fig. 4). Among the latter two bacterial classes, *Planctomycetacia* were predominantly found at the uninhabited island constituting on average 17% of the sequences as opposed to 8% at the inhabited island, whereas

*Deltaproteobacteria* comprised about 10% at the inhabited island compared to 6% at the uninhabited island. Water column bacterial communities, both free-living and particle-attached, also consisted mostly of *Gammaproteobacteria*, *Alphaproteobacteria*, and *Flavobacteriia* (Fig. 4). In the particle-attached fraction, *Cyanobacteria* further constituted a large proportion of sequences of on average 43% at the uninhabited and 37% at the inhabited island. Among these dominant bacterial classes, two and eight times higher proportions of *Alphaproteobacteria* and *Flavobacteriia*, respectively, were detected at the inhabited island in the free-living fraction. A similar trend was observed in particle-attached communities, although far less pronounced (Fig. 4).

At OTU-level resolution, distinct bacterial communities were observed within each of the different sampled habitats (ANOSIM, $R = 0.91$, $p = 0.001$; Fig. S3), with the strongest separation between sediment and water column communities. NMDS plots based on Bray–Curtis dissimilarities of bacterial communities from each of the sampled habitats further showed that samples from the inhabited and uninhabited island tended to cluster apart from each other (Fig. 5). These differences in community composition between the inhabited and uninhabited island were highest for sediment bacterial communities (ANOSIM, $R = 0.67$, $p = 0.001$), with average Bray–Curtis-dissimilarities of 55–60% at each island and 70% between the two islands. Within the free-living and particle-attached fraction, bacterial communities from the inhabited and uninhabited island were not well separated, and this separation was only found to be significant for the particle-attached communities due to the lower sample size for free-living communities at the uninhabited island (Free-living: ANOSIM, $R = 0.25$, $p = 0.087$; Particle-attached: ANOSIM, $R = 0.26$, $p = 0.002$). Average Bray–Curtis-dissimilarities were higher at the inhabited island for both free-living (64% compared to 55%) and particle-attached (64% compared to 44%) bacterial communities (Fig. S3A). Apart from the inhabitation status of the islands, patterns in bacterial community composition, as depicted in the NMDS plots, appeared to be strongly correlated with observed water quality parameters, predominantly chlorophyll *a*, inorganic phosphate, and TEP (Fig. 5).

Redundancy analyses confirmed that the inhabitation status of the islands had highest explanatory power for sediment bacterial communities (RDA, $R^2 = 0.18$, $F_{1,14} = 4.382$, $p < 0.001$; Table 2). The amount of variation in bacterial community composition explained by inhabitation status for the free-living and particle-attached fraction was considerably lower, although still statistically significant (RDA, $R^2 = 0.10$, $F_{1,10} = 2.230$, $p = 0.011$ and $R^2 = 0.10$, $F_{1,14} = 2.619$, $p = 0.001$, respectively; Table 2). The measured water quality parameters were often equally or better suited to explain community variability within the habitats (Table 2). Chlorophyll *a* combined with inorganic phosphate accounted for almost 30% of the variability in free-living communities, with each parameter individually contributing about 17%, and slightly improving the model fit compared to using inhabitation status as predictor (Table 2). For the particle-attached fraction, chlorophyll *a* in combination with TEP explained approximately 15% of the variability in community composition, whereas for sediment communities, chlorophyll *a*

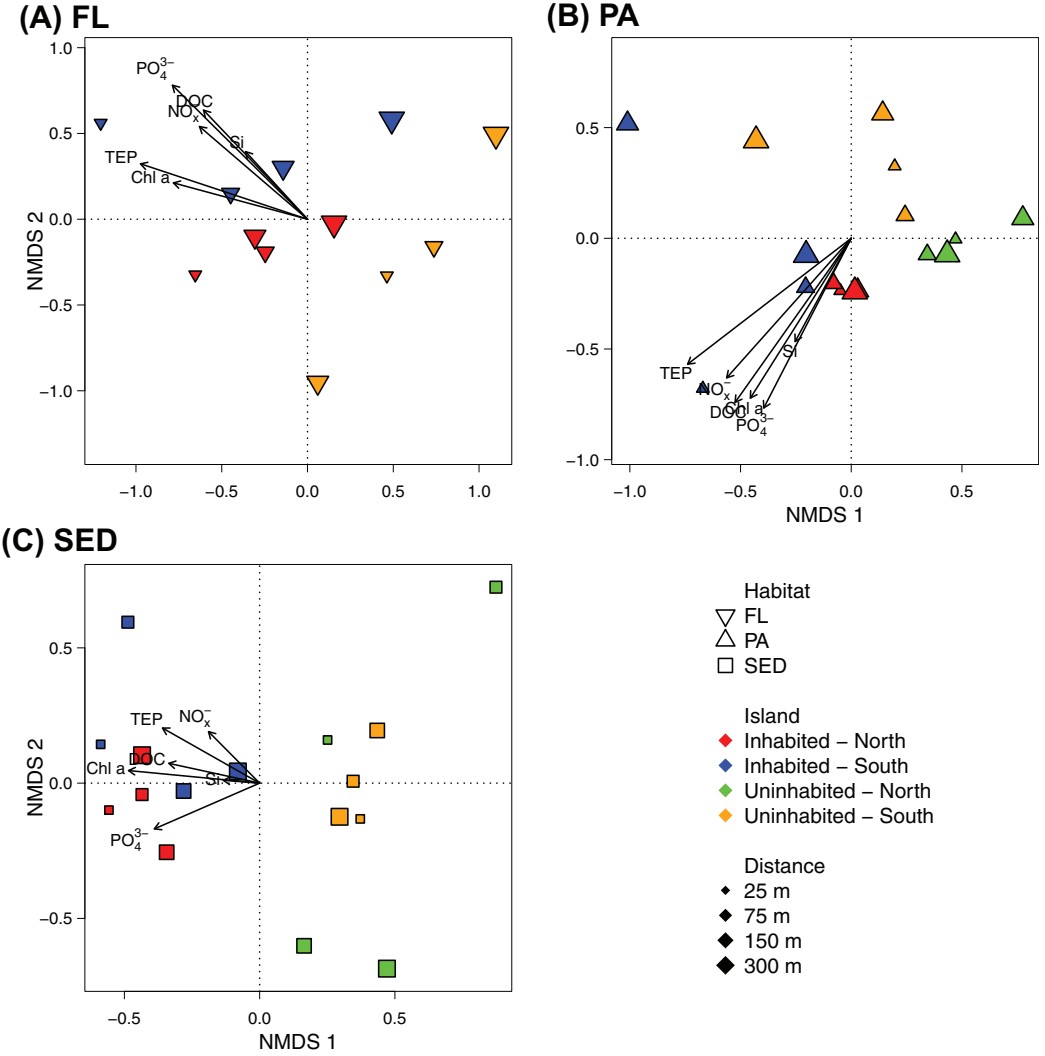

**Figure 5 NMDS plot of the bacterial communities of each sampled habitat at the inhabited island and uninhabited island.** (A) Free-living bacterial communities of the water column (>0.2 μm; FL), 2D stress: 0.11. (B) Particle-attached bacterial communities of the water column (>3 μm; PA), 2D-stress; 0.07. (C) Bacterial communities in reef sediments (SED), 2D-stress: 0.10. Arrows indicate *envfit* correlations of the water quality parameters with bacterial community composition. $NO_x^-$, nitrite/nitrate; $PO_4^{3-}$, phosphate; Si, silicate; Chl *a*, chlorophyll *a*; DOC, dissolved organic carbon; TEP, transparent exopolymer particles.

was as good a predictor as inhabitation status (RDA, $R^2 = 0.16$, $F_{1,14} = 3.979$, $p < 0.001$; Table 2).

We applied random forest models to the bacterial communities within the individual habitats to identify which OTUs were best suited to distinguish the inhabitation status of the islands. In general, the random forest models performed very well with an estimated out-of-bag error of less than 0.1, corresponding to one misclassified sample (Table S4). OTUs most important for the classification of inhabitation status were selected at a mean decrease in accuracy of more than 0.0020, 0.0030, and 0.0016 for free-living, particle-attached, and sediment communities, respectively (Table S4). Within the free-living

**Table 2 Contribution of environmental factors (inhabitation status and water quality parameters) to explaining variation in bacterial community composition in each of the sampled habitats based on redundancy analysis (RDA).**

| Habitat | Explanatory variable | AIC | Adjusted $R^2$ | F | df | p-Value |
|---|---|---|---|---|---|---|
| Free-living | | | | | | |
| | Inhabitation status | 98.94 | 10.1% | 2.230 | 1, 10 | **0.011** |
| | Chl $a$ + PO$_4^{3-}$ | 96.86 | 28.9% | 3.233 | 2, 9 | **0.001** |
| | Chl $a$ (pure) | | 17.1% | 3.407 | 1, 9 | **0.001** |
| | PO$_4^{3-}$ (pure) | | 16.8% | 3.364 | 1, 9 | **0.001** |
| Particle-attached | | | | | | |
| | Inhabitation status | 136.61 | 9.7% | 2.619 | 1, 14 | **0.001** |
| | Chl $a$ + TEP | 136.59 | 14.3% | 2.254 | 2, 13 | **0.001** |
| | Chl $a$ (pure) | | 6.0% | 1.980 | 1, 13 | **0.003** |
| | TEP (pure) | | 6.7% | 2.099 | 1, 13 | **0.013** |
| Sediment | | | | | | |
| | Inhabitation status | 140.86 | 17.7% | 4.221 | 1, 14 | **0.001** |
| | Chl $a$ | 141.07 | 16.6% | 3.979 | 1, 14 | **0.001** |

Notes:
p-Values defined as significant at a threshold of 0.05 are highlighted in bold.
Akaike information criterion (AIC) and adjusted $R^2$ are provided as goodness-of-fit metrics.
df, degrees of freedom (numerator, denominator); PO$_4^{3-}$, phosphate; Chl $a$, chlorophyll $a$; TEP, transparent exopolymer particles.

fraction, especially OTUs of the *NS5 marine group* of the *Flavobacteriia* were enriched at the inhabited island, as well as the gammaproteobacterial genus *Litoricola* and the alphaproteobacterial *PS1 clade* (Fig. S4). Among the depleted groups at the inhabited island were mainly taxa from the class *Gammaproteobacteria*, although from different genera as those showing enrichment, namely *Alcanivorax*, *Pseudoalteromonas*, and the *ZD0417 marine group*. In the particle-attached fraction mainly taxa from the classes *Gammaproteobacteria* (e.g., *Alteromonadaceae*, *Marinobacterium*, and *Litoricola*) and *Flavobacteriia* (*Flavobacteriaceae* and *Cryomorphaceae*) were enriched at the inhabited island (Fig. S4). Only one genus of the *Flavobacteriia*, *Algitalea*, was depleted in the particle-attached size fraction at the inhabited island. There were no taxa exhibiting a comparably strong enrichment or depletion in the sediment communities, since those communities displayed a more even OTU distribution. Nevertheless, pronounced differences in predominantly OTUs of the genera *Bythopirellula*, *Rhodopirellula*, and *Blastopirellula* (*Planctomycetacia*), which were depleted at the inhabited island, were observed. In contrast, *Desulfobulbaceae* and other deltaproteobacterial taxa, as well as *Robiginitalea* and *Actibacter* of the *Flavobacteriia*, were enriched at the inhabited island. This resulted in a well-supported classification of inhabitation status in the random forest model (Fig. S4). These OTUs with the highest importance in the random forest models contributed substantially to the bacterial communities in each habitat at the inhabited island, in some samples comprising more than 40%, 60%, and 8% each of the sequences in the free-living, particle-attached and sediment communities, respectively.

## DISCUSSION

### Inhabitation status determines water quality

Our results indicate that coastal waters around the inhabited island are significantly affected by anthropogenic inorganic and organic nutrient input. In the absence of other nutrient sources, e.g., rivers or surface runoff, a likely source of inorganic nutrient input is untreated sewage outflow that enters the reef waters through diffusive groundwater seepage (*D'Elia, Webb & Porter, 1981*; *Hwang, Lee & Kim, 2005*). This can lead to elevated concentrations of inorganic nutrients and DOC in the near-shore areas compared to sampling sites further out on the shelf, as was observed in this study as well as at other, larger islands (*Umezawa et al., 2002*; *Kim, 2003*; *Street et al., 2008*). Those observations of declining concentrations (with the exception of chlorophyll *a*) from sampling sites very close to the island to the ones further out on the reef flat, resemble patterns, and concentrations also observed at several other tropical and subtropical reef locations in the Hawaiian Archipelago, USA (*Laws, Brown & Peace, 2004*) and in Moorea, French Polynesia (*Nelson et al., 2011*). Declining concentrations of organic and inorganic nutrients are often related to biological and physical removal processes including dilution within the back-reef. Biological removal processes usually consist of the rapid assimilation of inorganic nutrients by benthic macroalgae (*Koop et al., 2001*; *Lapointe, Barile & Matzie, 2004*) and phytoplankton (*Hallock & Schlager, 1986*). Dissolved organic matter (DOM) released by phytoplankton, if not taken up by bacteria, often aggregates into larger molecules, such as TEP and ultimately particulate organic matter (*Passow & Alldredge, 1994*; *Verdugo et al., 2004*) and subsequently settles to the sediment, where it is metabolized by heterotrophic bacteria (*Wild et al., 2004*). Therefore, in addition to physically smothering benthic organisms by the amount of settling particles rich in organic matter (*Fabricius et al., 2003*; *Golbuu et al., 2008*) it can also harm them via a decreased oxygen availability (*Kline et al., 2006*) and a shallowing of the oxic sediment layers (*Brocke et al., 2015*).

In this study, apart from dilution, a large proportion of the introduced inorganic and organic nutrients is likely assimilated, e.g., by phytoplankton, benthic algae or seagrass, and transformed into the particulate fraction or metabolized close to the inhabited island, as measured concentrations decreased across the back-reef gradients. Such a rapid depletion across back-reef areas has been previously described for DOC, in that case by bacteria (*Nelson et al., 2011*). Furthermore, an immediate uptake of available inorganic nutrients by phytoplankton likely led to the observed differences in chlorophyll *a* concentrations, which were much more prominent than the differences in inorganic nutrient availability between the uninhabited and inhabited island (*Furnas et al., 2005*). The quantification of inorganic and organic nutrients may therefore not always be the most suitable indicator for anthropogenic eutrophication, unless they are combined with measurements of biological indicators, such as chlorophyll *a* concentrations or microbial community composition (*Paerl et al., 2003a*, *2003b*). It is also noteworthy that during high tide concentrations of chlorophyll *a* and inorganic nutrients are often lower than during low tide (*Welch & Isaac, 1967*; *Szmant & Forrester, 1996*). Thus, the concentrations

reported in this study are likely underestimating the terrestrial input compared to low tide measurements.

## Bacterial community composition: differences between habitats and islands

Taking all samples into account, habitat was the main factor determining community composition. This is supported by many studies from tropical coral reef ecosystems, including the Spermonde Archipelago, which showed distinct bacterial communities between various habitats within the reefs (*Rohwer et al., 2002*; *Tout et al., 2014*; *Polónia et al., 2015*; *Cleary et al., 2015*; *Kegler et al., 2017*). Different habitats vary considerably in substrate heterogeneity, colonizable surfaces, and food availability (*Bourne & Webster, 2013*), and will favor different bacterial life strategies (*Lauro et al., 2009*). Thus, investigations trying to disentangle the driving forces of community variability along gradients of water quality parameters should focus on the within-habitat differences of bacterial community composition. Indeed, while we detected differences in bacterial community composition correlated to changes in water-quality parameters in all investigated habitats, the response of different OTUs varied in each habitat. Therefore, we suggest separating not only sediment and water column bacterial communities, but also distinguishing free-living and particle-attached fractions within the water column in the analysis, especially in anthropogenically influenced coastal settings.

Interestingly, we observed opposite trends in the explanatory power of water quality parameters and inhabitation status from free-living to particle-attached to sediment bacterial communities. While inhabitation status on its own was not well suited to explain patterns in community composition within the free-living fraction, water quality parameters correlated strongly. Conversely, differences in sediment bacterial communities were best described by inhabitation status or chlorophyll *a* concentration, the water quality parameter with the most pronounced differences between the inhabited and uninhabited island. These observed differences in bacterial community response to changes in water quality may be subsequently attributed to individual taxon responses.

In this study, several OTUs from the *Flavobacteriia* were enriched in the free-living and particle-attached fractions of the water column at the inhabited island compared to the same habitats at the uninhabited island. Within the free-living fraction of the water column the *NS5 marine group* of *Flavobacteriia* were previously described to significantly correlate with phytoplankton blooms of *Akashiwo sanguine* followed by high abundances of *Litoricola* (*Yang et al., 2015*), which was also detected here in higher proportions at the inhabited island. *Flavobacteriia*, the largest class within the phylum *Bacteroidetes*, are key players in the initial degradation of the high molecular mass fraction of organic matter derived from algae production and detritus (*Kirchman, 2002*; *Pinhassi et al., 2004*). In the back-reef area of the inhabited island, chlorophyll *a* as proxy for phytoplankton abundance was consistently higher than at the uninhabited island. This may explain the higher proportion of *Flavobacteriia*, and within that class the observed *NS5 marine group*, and is consistent with observations of other studies (*Pinhassi et al., 2004*; *Williams et al., 2013*). After the initial degradation of the high molecular mass DOM, more labile DOM

is available for bacterial groups specialized in the uptake in lower molecular weight polysaccharides and other more labile phytoplankton exudates, such as *SAR11* and *Rhodobacterales* (both *Alphaproteobacteria*) and *Gammaproteobacteria* (*Morris, Frazar & Carlson, 2012*; *Williams et al., 2013*), of which especially the latter were often enriched at sampling stations high in nutrients or organic matter. Thus, they can be considered the start of the chain of DOM degradation fueling the microbial loop (*Azam et al., 1983*; *Fenchel, 2008*).

Particle-associated bacterial communities were further mainly enriched in OTUs belonging to the family *Alteromonadaceae* and the genus *Marinobacterium*, both members of the class *Gammaproteobacteria*. They feature a diverse repertoire of extracellular, hydrolytic enzymes, which enable them to access and assimilate various forms of organic matter on the aggregates in the water column or in the sediments (*Azam & Malfatti, 2007*; *Edwards et al., 2010*). Therefore, the observed higher diversity at the inhabited compared to the uninhabited island may be explained by OTUs from taxa being well adapted to using various forms of organic matter. *Gammaproteobacteria* is also a class which contains many strains of potentially pathogenic bacteria, such as *Vibrio* spp., and aggregates are known to be a hot-spot for pathogens (*Dinsdale et al., 2008*; *Garren et al., 2009*; *Lyons et al., 2010*). Particles with a higher abundance of potentially pathogenic bacteria can act as vectors for diseases from the water column to the sediments or via ingestion to higher trophic levels (*Lyons et al., 2005*). Pathogens already identified and found to be significantly enriched on aggregates in shallow coastal areas include the genera *Vibrio* and *Mycobacteria*, as well as the fecal indicator bacterium *Escherichia coli* (*Lyons et al., 2007*, *2010*). Notably, no correlation of the proportion of potentially pathogenic taxa with water quality has been previously reported for the Spermonde Archipelago on a large scale, although the prevalence of potential pathogens was generally high (*Kegler et al., 2017*). Here, we observed a comparable pattern, detecting potentially pathogenic taxa, especially the genus *Vibrio*, in similar proportions at both the inhabited and the uninhabited island, although their total sequence contribution did not exceed 4% (Fig. S5). Therefore, although we cannot confirm the pathogenicity of individual OTUs in this study, and *Gammaproteobacteria* and *Vibrio sp.* are also naturally occurring in healthy ecosystems, close monitoring, and further identification may help to avoid disease outbreaks.

Among the taxa enriched in the sediment samples at the anthropogenically impacted island, were, apart from the previously mentioned *Flavobacteriia* and *Gammaproteobacteria*, OTUs of the family *Desulfobulbaceae*. In organic matter-rich sediments, oxygen is quickly consumed in the upper, well-aerated layer of the sediment, and replaced by sulfate as electron acceptor in the degradation of organic matter, thereby leading to the expansion of hydrogen sulfide-rich layers, which are toxic to oxygen-consuming organisms (*Nielsen et al., 2010*). *Desulfobulbaceae*, also referred to as cable bacteria, are capable of preventing this expansion through a coupling of sulfide oxidation in anoxic sediment layers to oxygen reduction in oxic layers, spanning up to 1.5 cm (*Pfeffer et al., 2012*; *Reguera, 2012*). There, *Desulfobulbaceae* can prevent the expansion of sediment layers enriched in hydrogen sulfide and thus mitigate the effects of high organic matter loading. Although we did not measure hydrogen sulfide concentration or the

oxygen penetration depth of the sediment, the occurrence of these bacteria, together with other potentially sulfate-reducing taxa also found in this study (Fig. S3, e.g., *Desulfosarcina*, *Desulfofustis* or *Desulfococcus*), may indicate an increased availability of hydrogen sulfide in the deeper sediment layers at the inhabited island. In general, the occurrence of sulfate-reducing bacteria in reef sediments is common in the Indo-Pacific (*Hassenrück et al., 2016*; *Cleary et al., 2017*). Additionally, predictions are that with increasing urbanization of coastal areas, an increased release of nutrients and subsequent settlement of particles rich in organic matter, anaerobic sulfate-reducing bacteria will benefit from shallower oxic layers (*Brocke et al., 2015*). This could also affect coral health, as sulfate-reducing bacteria are an integral part of the microbial consortium responsible for black band disease in scleractinian corals (*Carlton & Richardson, 1995*; *Bourne, Muirhead & Sato, 2011*; *Sato et al., 2017*).

Overall, multiple lines of evidence suggest that small-scale changes in water quality between the inhabited and uninhabited island had strong effects on overall bacterial community composition. Especially TEP, phosphate, and chlorophyll *a* were highly correlated with differences in bacterial community composition between the inhabited and uninhabited island. Such island or reef-scale impacts of anthropogenic inorganic nutrient input are a novel finding for the Spermonde Archipelago, as local drivers of bacterial community variability were neglected in previous investigations. In that regard it might be an ideal model region to study the effects of even minute nutrient enrichments caused by local island populations on bacterial community composition. As almost all the islands are densely populated, equipping a few with basic sewage treatment facilities, e.g., septic tanks, could help to elucidate whether nutrient loadings and bacterial community composition around those "model islands" would resemble those of uninhabited islands of the region. This is of high relevance to regional management initiatives, especially since even such small changes often favor opportunistic, copiotrophic, and potentially pathogenic bacteria (*Eilers, Pernthaler & Amann, 2000*; *Dinsdale et al., 2008*). Therefore, future studies should additionally include a functional analysis of the activity of observed bacterial communities, such as RNA sequencing, to detect pathogenicity and identify the functional role of specific taxa in organic matter degradation.

## CONCLUSION

Overall, based on the differences between the uninhabited and the densely populated island, we infer that small island populations with limited wastewater treatment facilities can affect water quality in the surrounding back-reef area, which, in turn, shapes distinct bacterial communities. Most of the measured parameters were significantly elevated close to the inhabited island compared to samples from off the reef crest and the uninhabited island. This local eutrophication seems to be very limited compared to large-scale impacts of riverine and sewage input from the main land, e.g., in near-shore zones and estuaries (*Udy et al., 2005*; *Zhang et al., 2007*; *Reopanichkul et al., 2010*). However, even the relatively small inputs of inorganic and organic nutrients shape a bacterial community that is dominated by classes *Flavobacteriia* and *Gammaproteobacteria*, which

contain many opportunistic, copiotrophic bacterial taxa specialized in the degradation of organic matter. These shifts in bacterial community composition, in turn, could therefore also imply negative consequences for human or reef organism health. From a management perspective it appears (i) that those quickly responding bacterial communities may serve as priority bioindicators, and (ii) that sewage treatment is urgently required for these small islands.

## ACKNOWLEDGEMENTS

We would like to sincerely thank our Indonesian partners at Universitas Hasanuddin and the staff of the Barrang Lompo field station, and especially Nur Abu and Pak Ridwan, for technical support during the field work. We are further indebted to the Ministry of Research, Technology and Higher Education of the Republic of Indonesia (RISTEKDIKTI) for issuing the necessary permits. Our gratitude further goes to the technicians at the Leibniz Centre for Tropical Marine Research, Bremen, especially Matthias Birkicht and Dorothee Dasbach, for invaluable assistance with laboratory analyses.

### Funding

The authors received no funding for this work.

### Competing Interests

The authors declare that they have no competing interests.

### Author Contributions

- Hauke F. Kegler conceived and designed the experiments, performed the experiments, analyzed the data, prepared figures and/or tables, authored or reviewed drafts of the paper, approved the final draft.
- Christiane Hassenrück analyzed the data, prepared figures and/or tables, authored or reviewed drafts of the paper, approved the final draft.
- Pia Kegler performed the experiments, authored or reviewed drafts of the paper, approved the final draft.
- Tim C. Jennerjahn conceived and designed the experiments, analyzed the data, contributed reagents/materials/analysis tools, authored or reviewed drafts of the paper, approved the final draft.
- Muhammad Lukman conceived and designed the experiments, contributed reagents/materials/analysis tools, authored or reviewed drafts of the paper, approved the final draft, assisted in obtaining the necessary research permits.
- Jamaluddin Jompa conceived and designed the experiments, contributed reagents/materials/analysis tools, authored or reviewed drafts of the paper, approved the final draft, assisted in obtaining the necessary research permits.

- Astrid Gärdes conceived and designed the experiments, performed the experiments, analyzed the data, contributed reagents/materials/analysis tools, prepared figures and/or tables, authored or reviewed drafts of the paper, approved the final draft.

## Field Study Permissions

The following information was supplied relating to field study approvals (i.e., approving body and any reference numbers):

The required research permits to conduct fieldwork in Indonesia were provided by the Indonesian Ministry of Research, Technology, and Higher Education (RISTEKDIKTI).

## Data Availability

DNA-Sequence information is accessible at the European Nucleotide Archive (ENA) under accession number PRJEB18570.

## Supplemental Information

Supplemental information for this article can be found online at http://dx.doi.org/10.7717/peerj.4555#supplemental-information.

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
