# Peer review of "Small tropical islands with dense human population: differences in water quality of near-shore waters are associated with distinct bacterial communities"

_PeerJ, doi:10.7717/peerj.4555_

## Round 0.1 · original submission · Minor Revisions

Kegler et al. present a well-structured and carefully-conducted study of microbial community composition in the context of water quality in the coastal area of islands near Sulawesi, Indonesia. The manuscript was reviewed by two experts in the field, both of whom in my opinion provided thorough and high-quality reviews and recommendations for improvement of the manuscript. The reviewers agreed, and I concur, that the manuscript is of high quality and appropriate for publication in PeerJ following extensive revision following the guidance of the reviewers. While I have concurred with the reviewers that this is classified as "Minor Revisions" I do feel that the exceptional care taken by the reviewers should be noted and the authors should endeavor to substantially revise the manuscript following their excellent guidance.

I would add one final point: The word "result" in the title and the phrase "attributed to" (appearing twice) in the abstract are both inappropriate: causality cannot be determined from this study, and the approach of surveying two sites (albeit with several locations near each island) cannot allow the authors to attribute the differences in microbial community structure to the differences in water chemistry/biology. I recommend replacing "result in" in the title with the phrase "are associated with". I recommend replacing "can/could be attributed to" in the abstract with "are associated with" or similar.

Reviewer 1 ·

Basic reporting

The manuscript is well written, logically structured, and satisfies the requirements of an academic manuscript. The study and research questions are also self-contained and the results address the hypothesis that is posed within the introduction of the manuscript.
It is clear that the authors have made an effort to make their data accessible and their analysis reproducible. The raw data sequences are deposited within the European Nucleotides Archive and are available to download. Additionally, the metadata associated with these files is comprehensive and complete. The authors have also submitted well-annotated r-scripts that demonstrate how the amplicon data were processed, how statistical analysis of the data was completed, and how the map of the Spermonde archipelago was generated.

Below is a list of comments and suggestions that I have for the authors regarding basic reporting of the study that was completed.
1. Are there supplementary tables, figures, or documents? I am not able to see or review them, but they are referred to in the text of the manuscript (i.e. line 228, 239, 307).
2. Lines 228- 229, Line 234- 235: add “The” to the beginning of the sentence.
3. Line 223: remove the word “spectra” from this sentence and change the sentence to “Concentrations of the two different classes of molecules of organic matter, DOC and TEP . . .” or something similar.
4. Line 247: change “on” to “an”.
5. Lines 318- 323: restructure or break this sentence into two sentences for clarity.
6. Lines 323- 325: this statement should be supported with citations and belongs in the discussion section.
7. Lines 406- 407: insert “being” in between taxa and well and insert “using” between to and various.
8. Lines 405- 424: please restructure this discussion of Gammaproteobacteria abundance, Vibrio sp. presence, and pathogens. The content of this paragraph is important and of interest to the community, but the structure of this section is jumbled and harder to interpret than it needs to be. For example, state that Vibrio sp. was detected in samples from both the inhabited and uninhabited islands at similar relative abundances after line 417. Also, make it more apparent that even though Vibrio sp. have been implicated as pathogens in specific circumstances, the presence of Vibrio sp. genes does not necessarily indicate that the Vibrio sp. detected within these specific samples are pathogens. This is touched upon in lines 423-424, but a more obvious statement would serve this discussion.
9. Line 430: change “is” to “are”.
10. Line 435 and 437: change “can” to “may” to emphasize that these explanations are speculative and suggestive, not necessarily evidence-based.


Overall, the high quality figures and tables complement the content of the manuscript, but could be improved with the following suggestions:
1. Fig 2: replace the phrase “distance to island” on the x-axis with “distance from island”. The island is the origin of the x-axis so this phrasing logically makes more sense. Please rephrase this phrase when it is used elsewhere within the text and within other figures or figure captions.
2. Fig 4: label PA, SED, or FL next to each corresponding letter in the figure so the reader does not have to jump from the caption and then back to the figure.
3. Figure 5: Are there supposed to be “uninhabited North” data points included within plot A). These data are noticeably absent. As suggested above, label FL, PA, or SED next to each plot letter.
4. Figure 6: explain how these samples were grouped to generate this plot within the methods or underneath this figure in the caption (i.e. were all of the microbial samples grouped together- like PA collected from both transects at the inhabited island were combined into one category?)
5. Table 1: please condense the title for this table into one sentence and place the rest of the content within the method section. The parameter key should be placed underneath the table.
6. Table 2: see above comments on table title length.

Experimental design

The manuscript clearly outlines the research questions and discusses how the following research fills a knowledge gap, seeking to understand how inhabitation status of these small islands in Indonesia impact water quality parameters and microbial diversity and composition (of three different marine microbial community niches) along a distance gradient from the two islands.
The methods are described clearly and are reproducible to follow, especially because the authors have provided R scripts detailing the analysis of their data. Even so, please address the below comments.

1. Line 106-111: provide the depths of each sampling point if possible (especially because sediment samples were collected at each of these locations). Similarly, provide information on the benthic cover and/or the presence of reefs at these specific sampling points, as the presence of reefs can influence water quality and microbial community composition. This is also important because many connections are made between the presence of copiotrophic bacteria and how this functional group of bacteria may impact reef health within this manuscript.
2. Line 123: please provide what are the collection canisters made of and how were they cleaned prior to sample collection? These details should be included in the methods as they can impact the analysis (and can also introduce contamination to the samples).
3. Line 142-143: please clarify if this seawater was also collected in triplicate from the three different 5L canisters at teach transect X distance from each island.
4. Line 143: what is the rationalization between these size cut-offs?
5. Line 146: how were the sediment samples collected? Was it shallow enough for a snorkeler?
6. Line 150: separate “5” from “min” in “5min”.
7. Line 170: change “silvangs” to “SILVAngs”.
8. The sequence range for the PA, SED, and FL microbial community samples ranges from 964 – 49, 329 sequences per sample. While the topic of subsampling is controversial in our field, the differences in sequence number in this study may have influenced microbial community diversity amongst the samples and could contribute bias. Did the authors see a difference in the number of reads obtained from PA, SED, and FL samples that are correlated with the number of OTUs recovered from these samples? Please provide a comment regarding these potential biases and an argument as to why subsampling or other sequence standardization methods were not completed on the samples in this dataset.

Validity of the findings

1. Line 314: provide examples of which Gammaproteobacteria genera were depleted at the inhabited island.
2. The chla concentration in the inhabited North transect is initially twice the chla concentration in the inhabited south transect and the chla concentrations in the south transect of the inhabited island hover around ~ 5 ug/L. This is an interesting pattern that could be addressed or discussed.
3. Lines 433- 438: this is an interesting explanation, but needs to be expanded and linked back to the original research questions. Please provide taxonomic affiliations of the other “potentially sulfur-reducing taxa” that were recovered in this study and elaborate on the effects or impacts on high organic matter loading in tropical and shallow marine environments if possible. Have sulfur-reducers been recovered from coral reef sediment adjacent to highly populated areas? How does high organic matter loading tie into water quality of coral reef environments?
4. Lines 458- 465 and generally throughout the discussion- avoid overgeneralizations like using entire classes of bacteria to indicate disease prevalence or degraded marine habitat as a result of anthropogenic inputs and please rephrase these lines to mitigate this. For example, while specific genera within Gammaproteobacteria have been implicated as potential pathogens and/or with having copiotrophic lifestyles, not all OTUs within this class exhibit these behaviors. The reality is much more nuanced. This manuscript does not provide enough evidence to suggest that the Gammaproteobacteria recovered within these samples are pathogens (and this is stated once in line 423). Additionally, Gammaproteobacteria are normally present within the marine environment as well and this should be mentioned within the manuscript to provide a more comprehensive view of marine microbial diversity and composition.
5. Lines 445- 449: please elaborate upon how this region could be used in the future to study the effects of nutrient enrichment? Along these lines, please comment on if and how this study could be improved.

Additional comments

Nice work, overall!
Please remember to submit the supplemental files that are referred to within the manuscript. I am able to download a 'supplemental file' zip drive, but it is just a copy of the R-scripts that are already available.

Reviewer 2 ·

Basic reporting

The manuscript was generally well written and structured. I've provided some suggestions on rephrasing sections of the introduction and results (see comments to authors). There are several additional references that should be cited in the introduction, but the authors cited most of the appropriate literature. The objective were clearly stated.

Experimental design

This study provides an important contribution of the field of microbial ecology and highlights the potential role of anthropogenic inputs on altering biogeochemistry and ecosystem function. I do not have any issues with the experimental design. However, I do have a lot of questions about the statistical analyses (see comments to authors). Generally speaking, the authors need to provide more details on their analyses particularly in regards to how they present their mixed models.

Validity of the findings

With further clarification on their statistical approaches and interpretation, their results will be sound. The authors did a nice job of discussing their results in relation to previous research and providing possible explanations for patterns without over speculating the results.

Additional comments

Abstract-
Line 22- Since there have been a number of studies that have established gradients of water quality in Spermonde (e.g. Plass-Johnson et al. 2016, Polonia et al. 2015), I would rephrase this sentence to say something like “The present study is the first to investigate how bacterial community composition responds to water quality in several coral reef habitats, including the water column…..” Note, “back-reef” should be 2 words not hyphenated.

Line 30- This sentence needs to be reworded. Try something like this: “Bacterial communities in sediments are particle attached communities were significantly different between the two islands with bacterial taxa commonly associated with nutrient and organic matter rich conditions higher at the inhabited island.”

Line 37- I agree that Spermonde is a good model system for this type of question, but I think you can broaden the sentence. I would suggest something like: “Given the growing strain on coastal region, this study suggests that densely populated islands lacking sewage treatment have the ability to alter bacterial communities that may be important for normal coral reef ecosystem function.”

Introduction:
Line 44- There are definitely other classic studies that should be cited here (e.g. Lapointe et al. 2004 and other work done in Caribbean).
Line 46- this also leads to the proliferation of macroalgae

Line 53- I suggest starting a new paragraph when you discuss Spermonde. I would also suggest starting with a sentence that introduces water quality issues in Indonesia more holistically then segway into Spermonde as a model system.

Line 76. I think this paragraph should be placed after you intro paragraph before you start discussing Spermonde.

Line 91. Please review Lamb et al. 2017 (Seagrass ecosystems reduce exposure to bacterial pathogens of humans, fishes, and invertebrates) and cite here. This study assessed bacterial communities along gradients from shoreline to reef with and without seagrass beds at both of the islands you studied.

Line 101- What do you mean by “potential pathogen vector function” This sentence needs to be reworded.

Methods:
Line 108- what year were these surveys done?

Line 113- Why did you sample at high tide when the strongest land-based signal is typically found at low tide? In fact, you should mention in your discussion that because you only sampled at high tide, the water quality values you mentioned are likely underestimating terrestrial input.

Line 120- Was chla measured in the surface water?

Line 148- Were the samples extracted on Barrang Lompo or shipped somewhere else? How were they handled when shipped (e.g. dry ice, blue ice, etc)?

Line 180 I need further clarification on your GLMMs. The acronym GLMM is typically used in associated with generalized linear mixed models. You state that you ran “general linear mixed models”, which would make sense since you log-transformed your data presumably to meet assumptions of normality. If this is the case then you need to state why you log transformed your data. However, your use of permutation tests suggests that these are generalized linear mixed models, if this is the case what distribution did you fit to your data? With permutation tests, your F statistic is actually a pseudo-F statistic. You need to state more explicitly what your random and fixed effects are. Which package in R did you use to run your models? I’m assuming you ran separate models for each water quality parameter. If so, this needs to be stated here as well.

Line 202- What was your cut off for the VIFs in terms of collinearity? Which parameters were dropped? I would rephrase the last sentence to state that AIC were used to identify the best fit models and adjusted R2 values were used to determine goodness of fit of individual water quality paratmers.

Line 213- This should go at the beginning of the stats section

Results

Overall, the results need to be more concise. With your focus on the effects of water quality on community composition, what is the value of looking at bacterial diversity indices as well. If there is solid justification for including it then I would introduce diversity into your intro as well as state it in the objectives. Otherwise consider cutting it.

Lines 225-238, Table 1 and Figure 2: The GLMM (or LMM if you used linear mixed models) results need to be rewritten to be more concise, and better reflect the statistical analysis and results reported in Table 1. For example, it’s fine to start with significant patterns as a function of island and distance, but then you also need to mention that you found a significant interaction in several variables then go onto discuss underlying reasons for that interaction. I’m also confused why you are talking about transect level patterns when I presume you incorporated transect nested within island as your random effect? On a similar note, if you are treating transect nested within island as a random effect why are you presenting plotting individual transects in Figure 2.If you are truly interested in the between transect differences then you should include that as a fixed effect in the model. However, given that you seem to be most interested in island and distance from island I would suggest treating transect nested as a random effect.


Line 256- How were your confidence intervals generated (bootstrapping)? Were your “slightly stronger correlations” significant?

Line 277- this should be figure 5

Line 288- this should be fig. 5.

Line 291- I would argue that with a R2 of 0.18 island status has low explanatory power.


Discussion
Overall, the discussion was well written and summarized the results well.

Line 383- you need a transition sentence here. For example, “These differences in bacterial communities’ responses to water quality and proximity to human input may be attributed to individual taxon responses.”

---

## Round 0.2 · accepted · Accept

Thank you for addressing the various recommendations of the reviewers and from me. Your manuscript is now acceptable to be published in PeerJ.

Reviewer 1 ·

Basic reporting

no comment

Experimental design

no comment

Validity of the findings

no comment

Additional comments

The reviewers addressed my comments and I have no further suggestions for this manuscript. Nice work!